# Flowering Time Variation in Two Sympatric Tree Species Contributes to Avoid Competition for Pollinator Services

**DOI:** 10.3390/plants12193347

**Published:** 2023-09-22

**Authors:** Larissa Alves-de-Lima, Eduardo Soares Calixto, Marcos Lima de Oliveira, Letícia Rodrigues Novaes, Eduardo A. B. Almeida, Helena Maura Torezan-Silingardi

**Affiliations:** 1Instituto de Biologia, Universidade Federal de Uberlândia, Uberlândia 38400-902, MG, Brazil; 2Departamento de Biologia, Faculdade de Filosofia, Ciências e Letras, Universidade de São Paulo, Ribeirão Preto 14040-901, SP, Brazil; 3Entomology and Nematology Department, University of Florida, Gainesville, FL 3261, USA; 4Departamento de Biología Vegetal y Ecología, Universidad de Sevilla, 41009 Sevilla, Spain

**Keywords:** pollination, pollen, nectar, phenology, temporal partition

## Abstract

Competition is an important biological filter that can define crucial features of species’ natural history, like survival and reproduction success. We evaluated in the Brazilian tropical savanna whether two sympatric and congenereric species, *Qualea multiflora* Mart. and *Q. parviflora* Mart. (Vochysiaceae), compete for pollinator services, testing whether there is a better competitor or whether plants present any anti-competitive mechanism. Additionally, we investigated the breeding system, pollinators, and flowering phenology of both species. The results showed that *Q. multiflora* and *Q. parviflora* are dependent on pollinators for fruit formation, as they exhibited a self-incompatible and non-agamospermic breeding system. These plants shared the same guild of pollinators, which was formed by bees and hummingbirds, and an overlap in the flower visitation time was observed. Each plant species had different pollinator attraction strategies: *Q. multiflora* invested in floral resource quality, while *Q. parviflora* invested in resource quantity. The blooming time showed a temporal flowering partition, with highly sequential flowering and no overlap. *Qualea parviflora* bloomed intensely from September to October, while *Q. multiflora* bloomed from November to January, with the flowering peak occurring in December. The two *Qualea* species have morphologically similar flowers, are sympatric, and share the same pollinator community, with overlapping foraging activity during the day. However, they do not compete for pollinator services as they exhibit an anti-competitive mechanism mediated by temporal flowering partition.

## 1. Introduction

Environmental filters operate hierarchically in space and time and at different scales, shaping the community based on rules imposed for the establishment, permanence, and propagation of species [1]. For instance, evolutionary processes and historical events act as filters on global scales, while the selection of habitats and the dispersion capacity of species act on regional scales to determine community composition [2]. On a local scale, abiotic factors (such as precipitation, temperature, humidity, evapotranspiration, soil characteristics, and light availability) and biotic factors (biological interactions) are the main filters [2,3,4]. Regarding these biotic factors, pollination is an important mutualistic interaction, since many plant species rely on pollinators to produce fruits and seeds to establish and maintain viable populations in the environment [3,5,6,7]. This makes it a very important factor, especially for endangered species [8]. Plant communities are conditioned by many factors [3,8]. For example, environmental parameters such as temperature, humidity, and sunlight play a role in halophytic species [9] and savanna species [10]. Biotic interspecific interactions as local limitations in pollinator number and diversity also contribute to influencing plant communities [4]. Plant–pollinator interaction is based on animal behavior, considering the efficiency of resource collection, as well as on floral traits like morphology and phenology, which determine attractiveness, availability, and access to those resources [11,12,13].

Phylogenetically close plant species tend to share floral traits that influence the attractiveness of the same pollinator community [14,15]. Sympatric species with synchronous flowering can benefit from the sharing of the pollinator community by facilitating interactions, which increases attractiveness and consequently attracts a greater number of pollinators [16,17,18,19]. On the other hand, species with similar floral traits can be adversely affected by inter-species pollinator visits due to inadequate pollen deposition, also called heterospecific pollen transfer [20,21]. Heterospecific pollen transfer can reduce species’ fitness through the loss of pollen to heterospecific stigmas and the blocking of stigmatic surface to conspecific pollen [22,23]. In addition, simultaneous flowering of species with similar floral traits can lead to competition for pollinators, especially when pollinators show the same foraging behaviors between plant species, e.g., foraging at the same time and collecting the same floral resources [15,19,24], potentially decreasing the number of visits and the production of fruits and seeds [3,25,26,27]. Therefore, the results of the interaction between sympatric, phylogenetically close plants that share similar traits and floral resources, and share the same pollinator assemblage can be quite variable, ranging from facilitation to competition, thus shaping the plant community [3,28].

In systems where plants compete for pollinators, competition can be driven by the limitation of pollinators in the area or the pollinator’s preference for resources [25,29]. Pollinator preference for resources can involve both the quantity and the quality of the floral resources [30,31]. The number of flowers produced, along with their quantity of pollen grains and nectar, as well as the nectar’s sugar concentration, will influence the intensity of attraction to visitors [15,19,32,33,34,35], creating a positive relationship between the floral display and resources and the number of visits [36,37]. Thus, species that provide more resources, better quality resources, or resources that are available for more time in the environment can attract more pollinators and, therefore, be better competitors than others [34,38], accordingly excluding inferior competitors [39].

Notwithstanding the above, sympatric plant species with synchronous flowering can escape competition for pollinators by exhibiting fine adjustments in floral traits, as observed in the hummingbird-pollinated clade Iochrominae (Solanaceae) [40] and bee-pollinated *Dalechampia* species [41]. These adjustments can be determined by morphological distinctions (petal size, position of reproductive structures, and color spectrum) [42,43], or distinct phenological patterns such as the sequential flowering of species from the same family [44] or even the same genus [24], or the distinct moment of floral opening and releasing of floral resources such as pollen and nectar [12,45,46]. Furthermore, in the long term, differences might occur at the genetic level [47], which determines distinctions in the reproductive system (e.g., self-incompatible and self-compatible) [47,48]. Genetic changes combined with environmental factors can even affect the flowering period, leading to temporal niche specialization, which may be considered an anti-competitive mechanism [44,49,50,51]. In this context, one may expect that sympatric congener species may share the same pollinator assemblage and compete for their services [52] or that these species might have developed strategies to avoid this competition [15,19,24]. As each environment has its own abiotic and biotic characteristics, the study of these interactions, both competitive and anti-competitive, can increase our knowledge of how local ecological interactions influence ecosystem functioning. Such studies are particularly important in environments that are rapidly being degraded, as are tropical environments worldwide, such as savannas [53].

The most diversified savanna in the world is the Brazilian Cerrado, found in South America, with more than 7000 plant species, of which about 40% are endemic [54,55]. The Cerrado has lost approximately 46% of its natural areas due to fragmentation and less than 20% of its original vegetation is properly preserved [53]. The high proportion of endemic species together with the quick loss of natural areas makes Cerrado one of the hotspots for conservation in the world [56]. Studies of competitive interactions, considering how sympatric congener species compete or avoid such competition, and the relationships between abundant species and their pollinators are rare in Cerrado [57]. Vochysiaceae is a relevant plant family in the Cerrado, with high indices of importance values in many localities, and the genus *Qualea* presents very abundant species [58,59].

As competition is expected to occur between sympatric and phylogenetically close species, our main objective was to investigate whether trees from two *Qualea* species (Vochysiaceae) compete or avoid competition for pollinator services in a Cerrado area. To test whether they compete and, in this case, whether there is a better competitor, or whether there is an anti-competitive mechanism, we outlined a flowchart with four main specific objectives (Figure 1): (1) evaluate the reproductive system of both species, determining whether pollinators are essential for fruit formation; (2) if plants are pollinator-dependent, investigate if they share the pollinator community and if pollinators present the same foraging behavior to both species. Then, if we determined that these plants shared pollinators and that the pollinators were active at the same time for both species, we analyzed (3) which species is the best competitor at attracting more pollinators, and (4) whether these plant species may avoid competition by adopting a specific strategy.

## 2. Results

We found that *Q. multiflora* Mart. and *Q. parviflora* Mart. (1) depend on pollinators for fruit production, and as manual pollination generated more fruits than natural pollination, we can infer that the area may have lost part of its pollinator services. Both species (2) share the pollinator community, including the most effective pollinator species, which could lead to strong competition between these plant species. Additionally, *Q. multiflora* and *Q. parviflora* (3) offered different amounts of resources to pollinators, and both had (4) sequential flowering, which made them temporally able to avoid the competition for pollinators.

*Specific Objective 1*—Breeding System

*Qualea multiflora* and *Q. parviflora* are self-incompatible and have a non-agamospermic reproductive system since spontaneous self-pollination, manual self-pollination, and agamospermy treatments did not produce fruits (Table 1). Both species are pollinator-dependent for fruit production as outcrossing is essential. Interestingly, manual cross-pollination produced more fruits than natural pollination for both species. The ISI values indicate both species are allogamous and so, cross-pollination is mandatory. The REI values indicate that *Q. multiflora* receives a better amount of natural cross-pollination than *Q. parviflora.* The medium number of ovules and the fecundity rate were similar for both species.

*Specific Objective 2*—Pollinators

Flowers of *Q. multiflora* were visited 260 times by 13 pollinator species while flowers of *Q. parviflora* were visited 289 times by 14 species (Table 2). Pollinators of both species were made up of bees and hummingbirds. *Qualea multiflora* and *Q. parviflora* significantly shared the same pollinator community when considering either all potential pollinators (ANOSIM = −0.011; *p* = 0.518; stress = 0.154; Appendix A) or only effective pollinators (ANOSIM = −0.009; *p* = 0.514; stress = 0.176; Appendix A). They had twelve species in common, sharing all effective pollinators and three occasional pollinator species. Hummingbirds and less frequent bees were considered occasional pollinators.

Visits to *Q. multiflora* flowers began with daylight and just after anthesis around 05:00 a.m. and were distributed throughout the day. In *Q. parviflora*, visits started with the opening of the flowers at approximately 07:30 a.m. and showed two peaks of activity, from 08:00 to 09:30 h and 12:00 to 14:00 h. In both species, the end of insect activity was close to 16:00 h, but hummingbirds continued until sunset at about 18:30 h. All pollinator visits during the day between the two plant species overlap by 71.8–87.2% (Appendix A) and the effective pollinator by 74.9–87.7%, showing significant overlap patterns (*p* < 0.05; Figure 2, Appendix A).

*Specific Objective 3*—Plant Resources

Both species showed high availability of specific resources (however, see results of Specific Objective 4). *Qualea multiflora* produced significantly fewer inflorescences per individual plant and fewer open flowers per inflorescence, but higher amounts of pollen grains, nectar volume, and sugar concentration per day than *Q. parviflora* (Table 3, Appendix A). Nonetheless, both species showed a similar amount of viable pollen and floral buds per inflorescence (Table 3, Appendix A). Pollen release occurred earlier in *Q. multiflora* (05:00 h) than in *Q. parviflora* (07:30 h). Nectar was already available for collection before the floral opening in *Q. multiflora* (01:00 h) and *Q. parviflora* (06:00 h). On the second day, *Q. multiflora* flowers underwent changes in the bonding of their floral structures, but there were still viable pollen grains in the anther and a small amount of nectar inside the calcar. *Qualea parviflora* flowers provided resources for only one day, and at the end of that day, the flowers fell. During flower viability, both species exude sweet and light scent, with 33.3–56.5% overlap throughout the day, showing significant patterns of overlap (*p* < 0.003; Figure 3, Appendix A).

*Specific Objective 4*—Flowering phenology

Both species showed temporal flowering partition, with highly sequential flowering and no overlap. The flowering of *Q. multiflora* and *Q. parviflora* showed a significant difference in their activity peak (Watson two-test U^2^ = 1.721, *p* < 0.001; Figure 4a,b, Appendix A), with *Q. multiflora* peaking in December, shortly after the end of *Q. parviflora* flowering. Similarly, we observed no overlap of flowering between species, which was confirmed by the niche overlap null model (Figure 4c, Appendix A).

## 3. Discussion

*Qualea multiflora* and *Q. parviflora* flowering patterns are annual and unimodal with synchronism between individuals of the same species and population, as observed in other Brazilian savanna species [24,44,60]. Both species are allogamous and pollinator-dependent for fruit production, as indicated by the pollination experiments and the index of self-incompatibility (ISI). The fecundity rate indicates that both are able to turn ovules into seeds similarly. The larger REI value observed in *Q. multiflora* than in *Q. parviflora* may be a consequence of its pollination attraction, with higher amounts of pollen grains, nectar volume, and sugar concentration per day, all of them larger than observed in *Q. parviflora*, which can lead to longer visits and, consequently, a greater probability of promoting effective pollen exchange. In a study encompassing 100 angiosperm families, the authors noted that about 39% of the analyzed species exhibited the self-incompatible system [61]. Many savanna families [37,61,62], including the Vochysiaceae family [63,64,65,66], also exhibit the self-incompatible system and thus depend on pollination vectors to achieve cross-pollination, gamete fertilization, and consequent fruit production. The self-incompatible system is a common strategy for species pollinated by insects, as it favors the attraction of animals and facilitates the flow of pollen and cross-pollination [67].

The two studied species shared all pollinators and the effective pollinator community. A potential reason for this sharing of pollinators is that both species provide similar resources and have a similar floral display, which may be partially explained by their phylogenetic proximity [15,37,68]. The same floral resource offered continuously by congeneric species presenting sequential flowering permits the same guild of floral visitors to move from one blooming species to the next, facilitating the maintenance of pollinator populations in the area [10,15,19].

Finally, floral scent is an important characteristic that increases the number of floral visits as it helps the pollinator locate the flower [69,70]. Here, floral scent was easily assessed by humans, considered to be ‘odor blind’, as our perception is inferior compared to other animals’ perception [71]. Therefore, we suggest that future studies investigate the variation in the presence and quantity of the chemical components in floral scent throughout a flower’s life to enhance our understanding of scent’s relevance to pollinators.

Despite sharing the same pollinator assembly, presenting the same floral traits, co-occurring in the same area, and being congeners, these species of *Qualea* do not compete for pollinators due to sequential flowering, which can avoid direct competition and potentially optimize fruit production. The temporal niche partition observed between these savanna trees has already been observed in other species [15,19,37,68] and systems. In these studies, plant species demonstrate stable coexistence facilitated by temporal niche specialization in flowering time [39]. The coexistence of species within a community is determined by a series of processes collectively known as environmental filters [39,72]. The local species pool is shaped by regional species that undergo habitat selection, allowing them to disperse and coexist despite the pressures of interspecific interactions [73].

The temporal segregation of flowering may be a response to past competition for pollinator services [74]. Evolutionary pressure, such as competition for pollinators, may have favored plants with temporal flowering differentiation, reducing competition and maximizing plant reproductive success. It is expected that as species become more closely related, their niche axes overlap more, increasing the potential for competition [75]. The variation in flowering time among sympatric species that share the same pollinator community is fundamental for maintaining the reproductive success of many species. If these species were to flower simultaneously, they would be forced to share pollinators, which would be particularly detrimental to species, such as *Qualea*, that rely on pollen transfer for fruit production. The role of pollinators as the primary factor driving the evolutionary and ecological displacement of flowering time has been observed in plant species from different families. For example, a study observed that species with morphologically similar flowers, relying on nectar as their main resource and pollinated by a single species of hummingbird, exhibited different flowering times due to competition for pollinator services [76].

Sequential flowering in sympatric and congener species may result from various selective pressures at different stages of plant life or simply due to the influence of distinct environmental cues on flowering time. In the case of the species under study, as well as many other anemochoric species from the Cerrado, all life stages (such as newly flushed leaves, seed dispersal, and deciduousness) occur simultaneously, except for flowering [77,78]. Therefore, the temporal segregation in flowering between the two species is likely attributed to selective pressures acting during their flowering stages, rather than other pressures affecting different phenological stages. Abiotic variables such as precipitation, temperature, humidity, and day length may also affect important plant phenological processes, especially flowering. However, comparing the flowering period of these species in the study area against these abiotic factors revealed no significant effects [77,78]. Thus, our results together with these studies suggest that the temporal variation in flowering in these sympatric and congener species is potentially caused by selective pressure at the flowering stage.

While testing the factors responsible for sequential flowering in these *Qualea* species is challenging, we strongly believe that competition for pollinators is the most likely driver. Our tree species lack common anti-competitive strategies seen in sympatric plant species, such as a distinct reproductive system [48], diverse morphological characteristics [40,46], and separate daily periods of floral opening [45]. Sequential flowering not only benefits plant species by reducing competition for pollinators [79] but also serves as a facilitating strategy, with early-blooming species aiding pollinators in finding late-blooming species [80], thereby supporting overall pollinator populations. Similar flowering sequences have been observed in sympatric congener species from different families, such as Asteraceae, Lauraceae, Melastomataceae, Salicaceae [81], Myrtaceae [24], Malpighiaceae [44], and Fabaceae [82]. Flowering sequences are also observed among sympatric species of different genera within the same family. For instance, in a study conducted in the same area [15,44], sequential flowering with slight overlaps was observed in four species of Malpighiaceae, including three species of the genus *Banisteriopsis* and one of the genus *Peixotoa*. Malpighiaceae is a family with many predominantly pollinator-dependent species [15,44]. Therefore, this pattern of sequential flowering appears to be common among sympatric and phyllogenetically close species, and may be a result of competition for pollinators. Furthermore, the temporal flowering partition mechanism is an important factor that facilitates the increase and maintenance of species diversity [39]. Consequently, both *Q. multiflora* and *Q. parviflora* could be used together in reforestation programs as they are native Cerrado species easily found in its areas [57,83] and both are able to sustain the same guild of bees and hummingbirds consecutively.

## 4. Materials and Methods

### 4.1. Study Site

We conducted our study from September 2018 to March 2020 in a cerrado sensu stricto area (18°58′59″ S; 48°17′53″ W in the WGS84 coordinated system) within the ecological reserve of Clube Caça e Pesca Itororó de Uberlândia, in the state of Minas Gerais, central Brazil (Figure 5). Clube de Caça e Pesca Itororó covers a 640-hectare area west of the urban perimeter [84] and features vegetation consisting of 2–8 m high trees with an understory of shrubs and grasses [54]. The climate is classified as Aw according to the Köppen climate classification, which consists of two well-defined seasons: warm and wet from October to March (rainy season), and cold and dry from April to September (dry season), with a mean annual temperature of 22 °C and mean rainfall of 1.500 mm [85].

### 4.2. Plant Species

*Qualea multiflora* Mart. and *Q. parviflora* Mart. (Vochysiaceae) are native species of Brazil, widely distributed and easily found in the Amazon Forest, Atlantic Forest, Pantanal, Caatinga, and Cerrado areas of ‘campo rupestre’ and ‘cerrado strictu sensu’ [86]. They exhibit a shrub-tree habit with twisted branches and are highly recommended for the ecological restoration of degraded areas [83]. Their inflorescences are terminal and semi-terminal types, with flowers reduced to just one carpel, one stamen, one petal, and four sepals forming the calcar to store nectar [87,88]. Dried fruits are capsular with three locules and many seeds [89]. These species are among the most common in Cerrado and coexist in sympatry in any of its regions [90], including our study area [84]. In our study area, a floristic study of the tree species observed 33 plant families and 68 species, and identified the Vochisiaceae family as the most prominent, while *Q multiflora* and *Q. parviflora* exhibited the highest Importance Value Index [84]. In the present study, we gathered data from 65 individual plants of each species within the boundaries of the ecological reserve.

### 4.3. Data Collection and Analysis

We conducted four distinct experiments to assess the breeding system and reproductive success (Specific Objective 1), pollinators (Specific Objective 2), plant resources (Specific Objective 3), and flowering phenology (Specific Objective 4). Each experiment involved a separate set of plants. We selected conspecific individuals at least 10 m apart, all of which were growing under similar environmental conditions and displaying similar heights and branch numbers. *Qualea multiflora* trees measured 2–4 m in height, while *Q. parviflora* trees ranged from 2.5 to 9 m in height. Statistical analyses were performed using R 4.0.0 [91] with a 95% confidence level.

*Specific Objective 1*—Breeding System and Reproductive Success

To investigate whether the breeding system of both species is pollinator-dependent, we used 20 individuals per plant species. At the beginning of flowering, we tagged 30 floral buds per individual, which were equally divided into five treatments: (i) agamospermy, (ii) spontaneous self-pollination, (iii) manual self-pollination, (iv) manual cross-pollination, and (v) natural pollination, yielding 120 floral buds per treatment/species for a total of 1200 floral buds across both species. The agamospermy, manual self-pollination, and manual cross-pollination treatments had their floral buds previously bagged with mesh bags which were opened exclusively to perform these treatments. In agamospermy, we emasculated the buds in the pre-anthesis phase with partial removal of the sepals and the petals. In spontaneous self-pollination, floral buds were bagged, and pollination occurred without interference from any pollen vector, unlike manual self-pollination, in which shortly after the anthesis, we transferred the pollen grain manually from the anther to the stigma of the same flower touching the dehiscent anther and the stigma surface. We performed manual cross-pollination between flowers of individuals a minimum distance of 20 m apart. In the natural pollination treatment, we left the flowers free for any visitation. We bagged flowers with mesh bags as it allows light to enter and the exchange of gases, and protects flowers from contact with animals that can bring pollen grains. After performing the treatments, we bagged all the flowers again, except for the spontaneous self-pollination treatment, whose bags were not opened, and the natural pollination flowers, which were not bagged. Thirty days after pollination, we observed the produced fruits.

Reproductive success is related to many factors, such as the phenology of the interacting species, the presence and activity of pollinators, the index of self-incompatibility, the reproductive efficacy index, and the fecundity rate, which indicates if the species can maintain its population in the area [10]. To corroborate that plant species are pollinator-dependent, plants must not produce fruits by agamospermy or spontaneous self-pollination and must produce fruits by manual pollination. The manual self-pollination treatment determines the dependence or not of pollinators to transport the pollen to the stigma, and the cross-pollination treatment indicates whether the fruit production is dependent on the pollen from another individual of the same species. The natural pollination treatment acts as a control group. The index of self-incompatibility (ISI) indicates whether cross-pollination is required or not and is calculated by dividing the fruit set after self-pollination by the fruit set after cross-pollination [92]. The reproductive efficacy index (REI) assesses if the local pollinators are sufficient for adequate cross-pollination of the self-incompatible species. It is calculated by dividing the fruit set after natural pollination by the fruit set after manual cross-pollination [92]. The fecundity rate is determined by multiplying the seed/ovule and fruit/flower ratios [93], representing the percentage of ovules that develop into seeds. To calculate the mean number of ovules per flower, we examined ten flowers per species, each from a different plant.

*Specific Objective 2*—Pollinators

We used 15 individuals per plant species to investigate the pollinator community of each species to determine the degree to which they might overlap in species composition and foraging time behavior. We observed floral visitors on sunny days with low wind intensity, from 5:00 to 19:00 h. Preliminary assessments indicated that the frequency of floral visitors was much lower in the afternoon compared to the morning. Therefore, we concentrated our observations during the period of higher activity. Observation sessions lasted 45 min every hour, resulting in a total sampling effort of 40 h per plant species, with 30 h in the morning and 10 h in the afternoon. For each floral visitor, we recorded information on visit frequency, species richness, behavior, and foraging time. To assess the frequency of pollinators, we considered the flowers of the same individual as dependent samples. We identified vertebrates through direct observations and photographs. We collected at least one specimen of each insect species, which was identified and stored at the Laboratório de Ecologia Comportamental e de Interações of the Universidade Federal de Uberlândia, Brazil.

We classified floral visitors based on their behavior and visitation frequency in the field. A visitor was considered a pollinator if it regularly contacted both the stigma and the anther during flower visitation [41]. Effective pollinators were those that frequently interacted with and touched the anther and the stigma with high frequency at different times of the day and throughout the flowering period. Occasional pollinators, or secondary pollinators, visited flowers less frequently and occasionally touched the reproductive parts, resulting in lower pollination rates [94]. Climate changes, environmental disturbances caused by human activities, or diseases can impact the insect population differently, potentially leading to a decrease in sensitive species at the local level [95,96]. This can negatively affect pollination services. In such cases, less sensitive species may replace sensitive pollinators. However, it is important to note that our study area has not experienced significant disturbances, at least in the past fifteen years. Visitors who did not contribute to pollination were classified as non-pollinators and were excluded from the analysis.

To compare and verify whether these two plant species share the pollinator community, we performed two analyses: one considering all pollinators and the other considering only the effective pollinators. We performed these comparisons using an NMDS (non-metric multidimensional scaling) followed by the ANOSIM test (Analysis of Similarity) with 999 permutations and Euclidean distance for both analyses. These analyses were carried out with the ‘vegan’ package [97].

To assess the overlap of visits throughout the day, we considered the frequency of visits by pollinators on an hourly basis. We conducted this analysis for both the entire pollinator community and effective pollinators. To measure overlap, we used the overlap coefficient [98], Pianka and Czekanowski metrics, and null models of overlap on matrices of visit frequency over time. The overlap coefficient, which employs Kernel density estimates, quantifies overlap on a scale of 0 (no overlap) to 1 (total overlap). We used the Dhat1 estimator, particularly suited for small samples [99]. Null models were generated with two randomization algorithms (1000 randomizations each), RA3 and RA4, using the ‘EcoSimR’ package [100]. RA3 rearranges lines, while RA4 rearranges non-zero line values while maintaining the ‘niche breadth’ of each species.

*Specific Objective 3*—Plant Resources

To assess which species produce more pollen grains, more nectar, and nectar with the highest sugar content, we evaluated several factors, including the number of inflorescences, floral buds per inflorescence, and open flowers per day, as well as the quantity and quality of nutritional floral resources. Additionally, we monitored the intensity and availability of flower scent [101]. We quantified the inflorescences of 10 individuals per species. For these plants, we selected three inflorescences without open flowers per individual to determine the total number of flower buds. We monitored the same 30 inflorescences daily from the first flower opening until the end of the bloom to estimate the average number of open flowers per day.

To analyze nutritional floral resources, we bagged the floral buds with mesh bags to prevent floral visitors from collecting pollen and nectar. To determine the moment of pollen release, we observed 30 flowers equally distributed in 10 individuals per species. We estimated the release of pollen touching the single anther (both species have only one anther per flower) on a slightly rough, black surface. The moment of pollen release was determined when we observed the presence of yellow pollen grains on the black surface. To quantify the total number and viability of pollen, we used 10 flowers per species, one flower per individual. We collected the flowers after floral opening and stored them individually in falcon tubes placed in Styrofoam boxes with ice to prevent dehydration of the anther (adapted from [101]), and transported them to the laboratory. We counted the total pollen grains without blushing pollen, while for the pollen viability, we stained them with acetate carmine (adapted from [102]).

To determine the availability of nectar to visitors, we selected 30 floral buds per species, equally distributed among 10 individuals. Before observations, we cut the base of the flower spore to facilitate nectar assessment using a capillary tube with a volume of 1 µL. The spores were cut because their openings were very small, making it impossible to assess nectar in intact spores with capillary tubes. For the evaluation of nectar volume and sugar concentration, we used 21 newly opened flowers per species, distributed equally among seven individuals. We analyzed three flowers every 2 h, between 6:00 and 18:00 h, with one flower per individual. To measure nectar volume, we employed the same capillary tube (1 µL) used for assessing nectar availability, while sugar concentration (%Brix) was determined using a handheld refractometer (Eclipse model) [103]. These data were collected exclusively on the first day of flower opening for statistical analysis comparing nutritional resources. The statistical models are presented in Table 4, and all models were implemented using the ‘glmmTMB’ package [104].

We also evaluated the intensity of the floral scent, checking the presence and intensity of the floral scent from the floral opening until the flower falls, every hour [12,105]. We followed 15 flowers, equally distributed among five individuals per species. For each observation of the floral scent, we assigned the following scores: 0—no scent, 1—mild, 2—moderate, and 3—strong. These score assignments were performed by only one person (LAL) to not bias the data. To analyze whether there is an overlap of scent intensity during the day, we conducted overlap metrics (overlap coefficient, Pianka, Czekanowski, and niche overlap null models) as previously described in Specific Objective 2—Pollinators.

*Specific Objective 4*—Flowering Phenology

To assess whether both species exhibit different flowering peak activities, allowing them to avoid competition, we monitored 20 individuals per species with one observation per month from March/2019 to February/2020. We analyzed phenological synchrony using circular statistical analysis, considering the presence and absence of flowers [106]. Months were converted into angles (one month = 30°) and we used the abundance of individuals with flowers to estimate the mean vector (µ), the mean length of the vector (r), the median, the circular standard deviation, the Rayleigh test (Z), and (*p*) at a 5% probability [107]. We compared the maximum activity peak between species using the Watson–Williams test. To assess the unimodality of the data, we conducted Watson’s goodness test before performing circular analyses. To analyze the overlap of flowering, we used a temporal abundance matrix and applied overlap metrics (overlap coefficient, Pianka, Czekanowski) and niche overlap null models, as previously described in Specific Objective 2—Pollinators.

## 5. Conclusions

In our study, we observed that the sympatric species *Q. multiflora* and *Q. parviflora* have morphologically similar flowers, are allogamous, and share the same pollinator community, with foraging overlapping during the day. However, they do not compete for pollinator services due to temporal flowering partition. Our observations revealed that (1) they depend on pollinators for fruit formation because they exhibit a self-incompatible and non-agamospermic breeding system, (2) they share the same guild of pollinators, which visit the flowers at overlapping times, and (3) they share most floral traits, but (4) they exhibit a temporal flowering partition, characterized by highly sequential flowering with no overlap. Overall, our results suggest that sympatric congener species with high floral trait similarity and shared pollinator assemblages can avoid competition through a temporal flowering partition. The factors contributing to this temporal flowering partition, including competition for pollinators, warrant further investigation, as they play a crucial role in shaping ecosystems and diversity patterns. Future studies focusing on phenology, floral resources, and pollinators associated with *Qualea* species in different locations could provide valuable insights into this complex system. 

## Figures and Tables

**Figure 1 plants-12-03347-f001:**
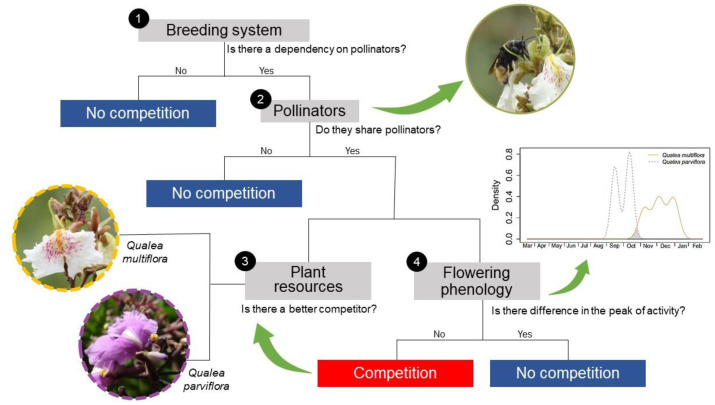
Flowchart to test whether sympatric congeners species, *Qualea multiflora* and *Q. parviflora*, compete for pollinators: Step 1: We assess whether both species depend on pollinators for fruiting, and if they do, we proceed to Step 2, where we observe the pollinators, determine whether the plant species share the same pollinator community, and assess whether the pollinators exhibit similar foraging behaviors. If there is a sharing of pollinators and time overlap during visitation, we analyzed two key aspects: in Step 3, we determine which species is the best competitor by analyzing the quantity and quality of the resources offered by both species, and in Step 4, we investigate whether these plant species avoid competition by adopting a temporal strategy, which would involve no overlapping of flowering peaks.

**Figure 2 plants-12-03347-f002:**
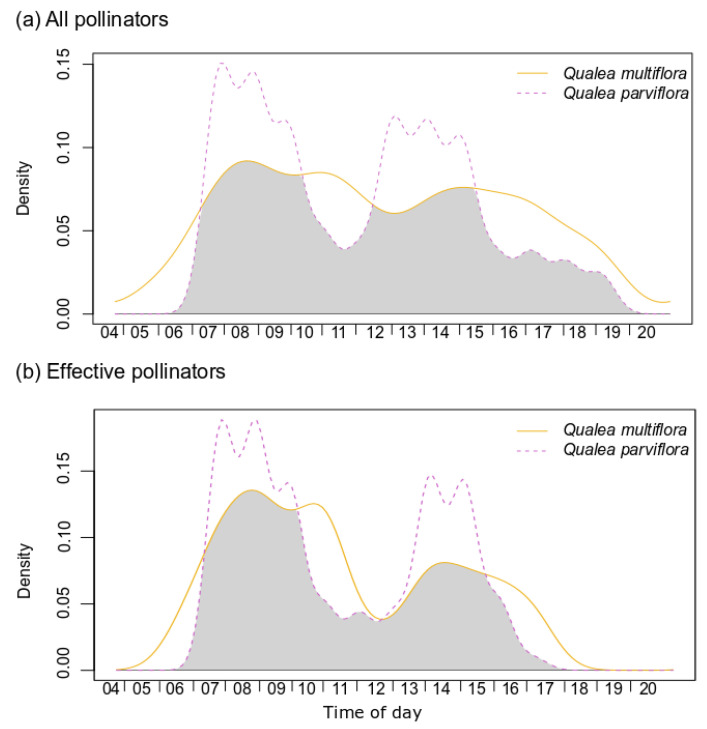
Kernel density functions of visits from all pollinators (**a**) and effective pollinators (**b**) throughout the day in *Qualea multiflora* and *Q. parviflora*. Shaded areas correspond to the overlap coefficient. Statistical results are depicted in Appendix A.

**Figure 3 plants-12-03347-f003:**
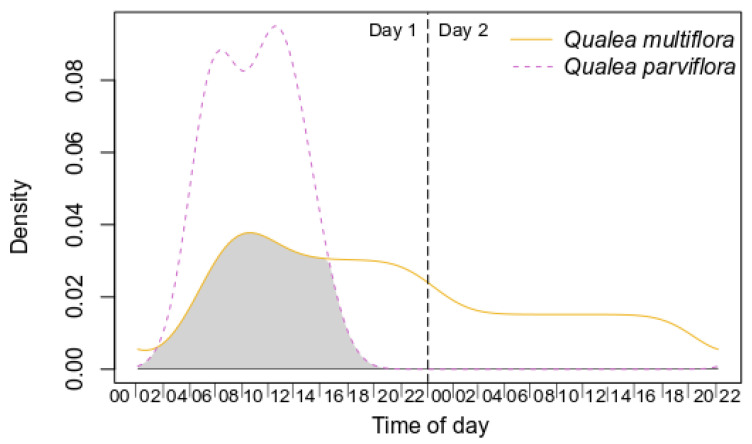
Kernel density functions for odor from pollinators from *Qualea multiflora* and *Q. parviflora*. Shaded areas correspond to the coefficient of overlap. Statistical results are depicted in Appendix A.

**Figure 4 plants-12-03347-f004:**
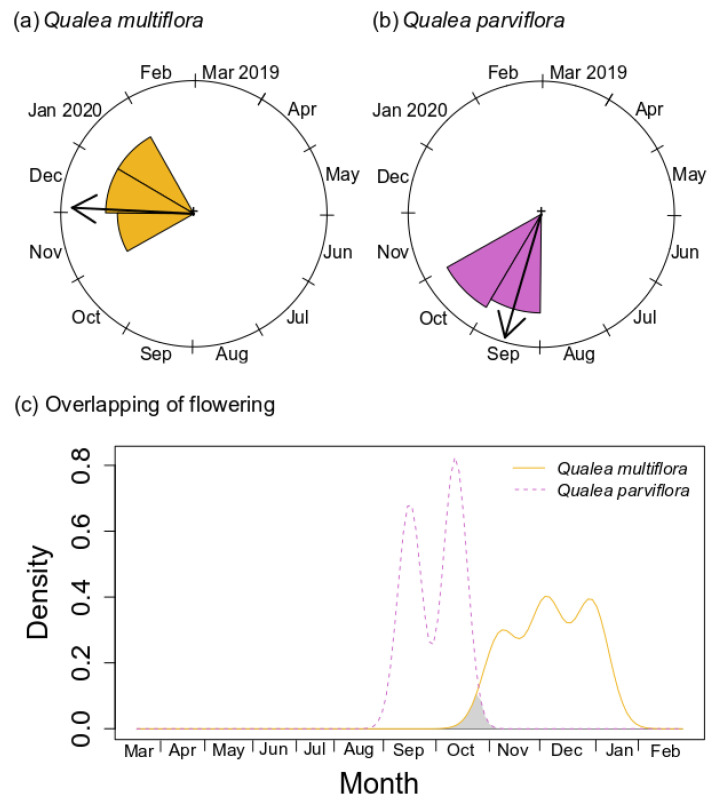
Histogram representing the flowering of *Qualea multiflora* (**a**) and *Q. parviflora* (**b**) from March/2019 to February/2020. Kernel density flowery from *Qualea multiflora* and *Q. parviflora*. Shaded areas correspond to the coefficient of overlap (**c**). Statistical results are depicted in Appendix A.

**Figure 5 plants-12-03347-f005:**
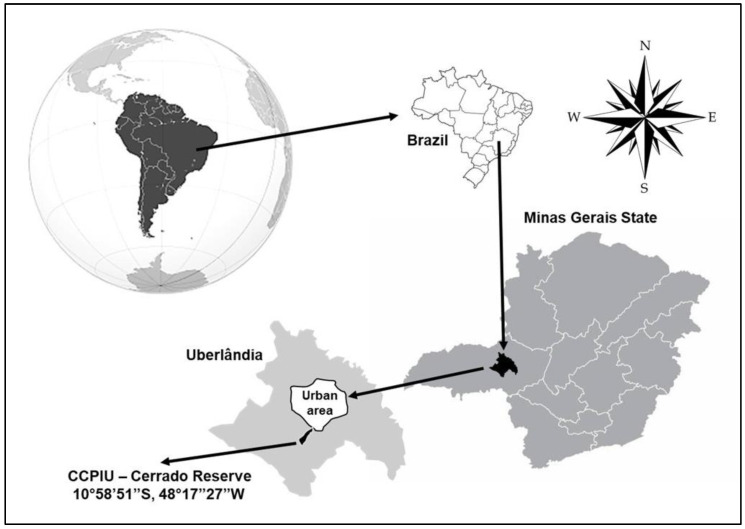
Localization of the ecological reserve of Clube Caça e Pesca Itororó near the urban area of Uberlândia city, in the state of Minas Gerais, central Brazil.

**Table 1 plants-12-03347-t001:** Pollination treatments and related values of *Qualea multiflora* and *Q. parviflora* (Vochysiaceae) obtained at the ecological reserve of Clube de Caça e Pesca Itororó de Uberlândia, Minas Gerais, Brazil.

Treatments	*Qualea multiflora*	*Qualea parviflora*
Agamospermy	0%	0%
Spontaneous self-pollination	0%	0%
Manual self-pollination	0%	0%
Manual cross-pollination	15.00%	8.33%
Natural pollination	12.50%	4.16%
ISI	0	0
REI	0.833	0.499
Medium number of ovules	12 ± 2.91	13.7 ± 3.88
Fecundity rate	0.85	0.81

**Table 2 plants-12-03347-t002:** Absolute (AF) and relative (RF) frequencies of effective and occasional pollinators of *Qualea multiflora* and *Q. parviflora* (Vochysiaceae).

Pollinators	*Qualea multiflora*		*Qualea parviflora*	
	AF	RF (%)	AF	RF (%)
Effective				
*Bombus (Fervidobombus) morio* (Swederus, 1787)	12	4.620	29	10.034
*Centris (Centris) aenea* Lepeletier, 1841	9	3.462	13	4.498
*Epicharis (Xanthepicaris) bicolor* Smith, 1854	25	9.615	13	4.498
*Epicharis (Epicharana) flava* Friese, 1900	15	5.770	9	3.114
*Epicharis (Epicharitides) cockerelli* Friese, 1900	12	4.615	15	5.190
*Epicharis (Triepicharis) analis* Lepeletier, 1841	7	2.692	16	5.536
*Paratrigona lineata* (Lepeletier, 1836)	30	11.538	15	5.190
*Xylocopa (Megaxylocopa) frontalis* (Olivier, 1789)	21	8.077	45	15.570
*Xylocopa (Neoxylocopa) suspecta* Moure and Camargo, 1988	45	17.308	51	17.647
Occasional				
*Amazilia fimbriata* (Gmelin, 1788)	32	12.307	21	7.266
*Centris (Aphemisia) mocsari* Friese, 1899	0	0	4	1.384
*Eufriesea auriceps* (Friese, 1899)	2	0.767	0	0
*Exomalopsis fulvofasciata* Smith, 1879	3	1.153	4	1.384
*Heliomaster squamosus* (Temminck, 1823)	47	18.076	54	18.685
Total	260	100	289	100

**Table 3 plants-12-03347-t003:** Availability of inflorescences, flower buds, and flowers and evaluation of the nutritional floral resources of *Qualea multiflora* and *Q. parviflora*.

	*Qualea multiflora* (Mean ± sd)	*Qualea parviflora* (Mean ± sd)	χ^2^	*p*
Inflorescences	18 ± 7.7	40 ± 17.6	79.15	<0.001
Floral buds	26.0 ± 10.4	27.0 ± 13.2	0.56	0.451
Flowers per day	1.2 ± 0.8	1.7 ± 1.1	22.13	<0.001
Pollen grains	9958.4 ± 344.8	9498.8 ± 49.4	121.50	<0.001
Viable pollen	9091.9 ± 762.2	8552.2 ± 58.8	0.49	0.484
Volume of nectar (µL)	0.77 ± 0.12	0.50 ± 0.29	16.17	<0.001
Sugar concentration (%Brix)	45 ± 12	32 ± 11	14.25	<0.001

**Table 4 plants-12-03347-t004:** Performed statistical models to compare floral resources and attractiveness between *Qualea multiflora* and *Q. parviflora*.

	N/Species	Model	Distribution	Variable Response	Fixed Predictor Variable	Random Predictor Variable
Inflorescences	10	GLM	Poisson	Inflorescence length	Species	-
Floral buds	30	GLMM	Poisson	Quantity of flower buds per inflorescence	Species	1|Individual/Inflorescence
Flowers per day	30	GLMM	Poisson	Number of flowers opened by inflorescence per day	Species	1|Individual/Inflorescences/Day
Pollen grains	10	GLM	Poisson	Total pollen grains per flower	Species	-
Volume of nectar (µL)	21	LMM	Gaussian	Volume of nectar	Species	1|Individual
Sugar concentration (%Brix)	21	GLMM	Beta	Percentage of sugar concentration in nectar	Species	1|Individual

## Data Availability

We will share the research data and make it available through Dryad.

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
