# Peer review of "Flowering Time Variation in Two Sympatric Tree Species Contributes to Avoid Competition for Pollinator Services"

_plants, 2023, doi:10.3390/plants12193347_

Round 1
Reviewer 1 Report
The authors propose a manuscript titled “Flowering‐time variation in two sympatric tree species contributes to avoid competition for pollinator services”. The article is original, well structured and written. In particular, this study takes into consideration and highlights an interisting starting to crucial concept Competition which can define crucial features of species natural history, like survival and reproduction success. The manuscript evaluated peculiar habitat located in the Brazilian tropical savanna two species: Qualea multiflora and Q. parviflora, that compete for pollinator services, testing whether there is a better competitor or whether plants present any anti-competitive mechanism. the study evaluated also the breeding system, pollinators, and flowering phenology of both species. The two plants considered shared the same guild of pollinators formed by bees and hummingbirds and there was an overlap in the flower visitation time. The two species have morphologically similar flowers, are sympatric, and share the same pollinator community with foraging overlapping during the day, however they do not compete for pollinator services as they show an anti-competitive mechanism mediated by temporal flowering partition. Very interesting data and approch that I share which I would summarize as "a return to the past with a modern approach".
I read the work carefully in a critical way, and suggested in detail some concepts only in order to further improve the good work done. In particular, some concepts need to be referenced, indicating point by point where is necessary.
In the work it is necessary to take into consideration and say something on crucial aspects.
1. Introduction
Well done, the concept are correct but in some case need to complete with literature data because they are not a study result or authors considerations. In bold the suggetsions
· Lines 32-44. On a local scale, abiotic (such as precipitation, temperature, humidity, evapotranspiration, soil characteristics, and light availability) and biotic (biological interactions) factors are the main filters [3,4]. Regarding these biotic factors, pollination is an important mutualistic interaction, since many vascular plant species rely on pollinators to produce fruits and seeds to maintain their population in the environment [3,5–7], and this is a very important factor especially for endangered species [Perrino et al. 2022], ultimately shaping plant communities [8], which differ according to the environmental parameters such as for example for halophytic species [Accogli et al. 2023]….please continue and complete the concept with other examples;
· Lines 53-54. In addition, simultaneous flowering of species with similar floral traits can lead to competition for pollinators (choose a reference)”;
· Lines 63-64. Pollinator preference for resources can involve both the quantity and the quality of the floral resources [choose a reference]”;
· Lines 71-72. Please some plant species examples on: sympatric plant species with synchronous flowering can escape competition for pollinators by exhibiting fine adjustments in floral traits;
· Lines 81-84. In this context, one may expect that sympatric congener species may share the same pollinator assembly and compete for their services [Pisani et al. 2021] or that these species might have strategies to avoid this competition [choose a reference];
· Figure 1: Well done
References to be added:
ü Perrino, E.V.; Tomaselli, V.; Wagensommer, R.P.; Silletti, G.N.; Esposito, A.; Stinca A. Ophioglossum lusitanicum L.: New Records of Plant Community and 92/43/EEC Habitat in Italy. Agronomy, 2022, 12, 3188. Doi: 10.3390/ agronomy12123188
ü Accogli, R.; Tomaselli, V.; Direnzo, P.; Perrino, E.V.; Albanese, G.; Urbano, M.; Laghetti, G. Edible Halophytes and Halo-Tolerant Species in Apulia Region (Southeastern Italy): Biogeography, Traditional Food Use and Potential Sustainable Crops. Plants, 2023, 12, 549. Doi: 10.3390/plants12030549
ü Pisani, D.; Pazienza, P.; Perrino, E.V.; Caporale, D.; De Lucia, C. The Economic Valuation of Ecosystem Services of Biodiversity Components in Protected Areas: A Review for a Framework of Analysis for the Gargano National Park. Sustainability, 2021, 13, 11726. doi: 10.3390/su132111726
2. Results and 3. Discussion
Some suggestions:
· Line 123. For botanical point of view is better to reporting for the first time in the paragraph the complete name of the species: Qualea multiflora …
· Figure 2. Please enlarge the size is too small;
· Well done, the tables 1 and 2 are clear;
· Some information on areal distribution of Qualea multiflora and Q. parviflora?;
· Line 221-223. Finally, floral scent is an important characteristic that increases the number of floral visits as it helps the pollinator to locate the flower [choose a reference]. Importantly, floral scent was done by humans, which is distinct from the insect perception [choose a reference]…
4. Materials and methods
Well done, only two suggetsions.
· I suggest to consider a map of the study area, yes I know will be too wide;
· Please specify the geographical system used for geographical coordinates (18°58’59” S; 48°17’53” W), WGS84?
5. Conclusions
Please spend two more words on future research in the field
Reviewer 2 Report
This manuscript titled “Flowering‐time variation in two sympatric tree species contributes to avoid competition for pollinator services” is an interesting topic discussing how two plant species have adopted different flowering strategies to avoid pollinator competition. To explain this, study was planned in four different experiments. However, I think this study lacks technical replicates due to which the outcome of this study cannot be consolidated. Please address following points,
1. I would highly recommend representing the results of “Specific Objective 1 – Breeding system” in a form of a diagram or graph as different strategies were adopted.
2. Please refer to line 140-141 “The REI observed for Q. multiflora (0,833) and Q. parviflora (0,499) indicate the first receives a better amount of natural cross-pollination than the second.”. what is the reason behind it as both are sympatric and phylogenetically close species?
3. How can you justify the “Specific Objective 2 – Pollinators” results as it does not consider any factor for pollinator population. For example, the pollinators you classified as “Occasional” is maybe due to its less population because of climate changes or any other environmental factors in study period/season but plays critical role in actual.
4. Current methodology adopted is not significant enough to conclude that Q. multiflora and Q. parviflora have adopted alternate flowering strategies in order to avoid competition as it lacks the technical replicates of treatments.
Moderate editing of English language is required.
Author Response
Dear rewier 2,
I hope this message finds you well.
We sincerely appreciate your valuable suggestions and improvements to the manuscript. Your expertise in this field is highly regarded, and your insights have been invaluable.
We have diligently worked to address your comments and incorporate the requested modifications and additions. You can easily identify all these changes in the new version of the manuscript highlighted in blue.
In this letter, we have taken the opportunity to respond to each of your questions and comments, also presented in blue for clarity.
Thank you once again for your time and thoughtful review.
Sincerely,
Helena Maura Torezan Silingardi.
Questions and comments followed by our responses:
- I would highly recommend representing the results of “Specific Objective 1 – Breeding system” in a form of a diagram or graph as different strategies were adopted.
Response: We modified the paragraph and organized the breeding systems results into Table 1 (lines 148-150).
- Please refer to line 140-141 “The REI observed for Q. multiflora(0,833) and Q. parviflora(0,499) indicate the first receives a better amount of natural cross-pollination than the second.”. what is the reason behind it as both are sympatric and phylogenetically close species?
Response: Although Q. multiflora and Q. parviflora are sympatric and phylogenetically close species, differences as the higher amounts of pollen grains, nectar volume and sugar concentration per day observed in Q. multiflora improve its attractiveness to pollinators. With more pollen and nectar to offer, floral visitors could stay longer in the flower to collect these resources and consequently promote a better pollen exchange, explaining the biggest REI value of Q. multiflora. We modified the first paragraph of the Discussion (lines 214-218) to include this explanation.
- How can you justify the “Specific Objective 2 – Pollinators” results as it does not consider any factor for pollinator population. For example, the pollinators you classified as “Occasional” is maybe due to its less population because of climate changes or any other environmental factors in study period/season but plays critical role in actual.
Response: We included more information on ‘Specific Objective 2 – Pollinators’ (lines 329-403) from references Hansen et al. (2000), Rosas-Gerreiro et al. (2014), Faurot-Daniels (2020) and González-Tokman (2020) to make our point clear about the definition of effective and occasional pollinators and the reasons species densities vary.
- Current methodology adopted is not significant enough to conclude that Q. multiflora and Q. parviflora have adopted alternate flowering strategies in order to avoid competition as it lacks the technical replicates of treatments.
Response: We understand your concern and appreciate your alert. However, we respectfully present a counterpoint. The combination of data we have collected using various methodologies, especially those focused on phenological aspects, allows us to suggest that these plants have adopted different flowering strategies. This methodology enables us to demonstrate that the blooming time exhibits temporal flowering partition, with highly sequential flowering and no overlap, as Q. parviflora bloomed intensely in September-October, and Q. multiflora bloomed from November to January, with the flowering peak in December. Additionally, we collected our data in a large area of approximately 640 hectares, encompassing our two sympatric and congeneric species. This area serves as an excellent ecological reserve with well-preserved cerrado vegetation, which has been utilized for numerous ecological studies. We have demonstrated that the species share the same floral morphology and floral resources, but pollen and nectar vary in quantity and quality. Both Qualea species rely on animal pollen transfer as cross-pollination is mandatory, performed by exactly the same group of bees and a very similar group of birds that collect floral resources during the same hours of the day. Although the plant species do not compete for pollinators since they bloom sequentially, as indicated by their phenology. Please see that we are not deterministic in our statements, as in the sentence (283-286): Thus, our results, together with these studies, suggest that the temporal variation of flowering in these sympatric and congeneric species is potentially caused by selective pressure at the flowering stage. And lines 494-495 - New studies including phenology, floral resources and pollinators associated to Qualea species in other localities could help to understand this complex system. Thank you for your comprehension.
Comments on the Quality of English Language: Moderate editing of English language is required.
Response: We have improved the language with the assistance of Professor Gregory Hocutt from Mesa Community College, Arizona, and believe it now meets the desired standard.
____________________________________________________________________________
Round 2
Reviewer 1 Report
Dear authors,
this last version of the manuscript sotisfy my requests. In my opinion the work is now able to be published on Plants journal without other changes.
Regards,
reviewer
Reviewer 2 Report
Thank you authors, I am satisfied with the revisions, the manuscript can be accepted in its present form.